# Anti-Stokes Photoluminescence in Halide Perovskite Nanocrystals: From Understanding the Mechanism towards Application in Fully Solid-State Optical Cooling

**DOI:** 10.3390/nano13121833

**Published:** 2023-06-09

**Authors:** Nikolay S. Pokryshkin, Vladimir N. Mantsevich, Victor Y. Timoshenko

**Affiliations:** 1Faculty of Physics, Lomonosov Moscow State University, 119991 Moscow, Russia; nspokryshkin@mephi.ru (N.S.P.);; 2Phys-Bio Institute, University “MEPhI”, 115409 Moscow, Russia; 3Lebedev Physical Institute, Russian Academy of Sciences, 119991 Moscow, Russia

**Keywords:** photoluminescence, anti-Stokes, halide perovskites, nanocrystals, phonons, optical cooling, laser cooling

## Abstract

Anti-Stokes photoluminescence (ASPL) is an up-conversion phonon-assisted process of radiative recombination of photoexcited charge carriers when the ASPL photon energy is above the excitation one. This process can be very efficient in nanocrystals (NCs) of metalorganic and inorganic semiconductors with perovskite (Pe) crystal structure. In this review, we present an analysis of the basic mechanisms of ASPL and discuss its efficiency depending on the size distribution and surface passivation of Pe-NCs as well as the optical excitation energy and temperature. When the ASPL process is sufficiently efficient, it can result in an escape of most of the optical excitation together with the phonon energy from the Pe-NCs. It can be used in optical fully solid-state cooling or optical refrigeration.

## 1. Introduction

Semiconductor bulk perovskite (Pe) materials and Pe-based low-dimensional structures such as nanocrystals (NCs) have attracted much attention as new optoelectronic materials with outstanding optical and electronic properties [1,2]. In recent years, halide Pe-semiconductors have been extensively studied for applications in optoelectronic devices, including solar cells and light-emitting diodes [3,4,5,6]. The tunable bandgap of Pe-NCs across the whole visible range [2,7] makes them favorable candidates in photovoltaics and photonics. The bright excitonic photoluminescence (PL) of Pe-NCs is promising for a new generation of light emitters with ultrahigh color purity [1,7,8].

Halide perovskites possess the crystal structure of ABX_3_-type, where A is an organic or inorganic cation, such as methylammonium (MA), formamidinium (FA), and cesium (Cs); B is a divalent metal cation (Pb, Sn and other); and X is a halide anion (Cl, Br, I or their mixture) as schematically shown in Figure 1a. The bandgap energies and PL spectra of Pe-NCs are determined by both their composition and sizes. The latter controls the band gap and PL properties of those NCs due to an effect of quantum confinement for charge carriers and excitons [7,8,9,10,11]. For example, colloidal CsPbX3 NCs are tunable in the spectral region of 410–700 nm by changing the anion content and nanocrystal sizes (Figure 1b). This is because the quantum size effect of the band gap of CsPbX3 NCs changes from about 2.4 to 2.7 eV when the mean NC sizes decrease from 11.8 to 2 nm, respectively [7]. PL spectra of Pe-NCs typically represent narrow emission lines with a linewidth of 12–42 nm, a wide color gamut covering up to 140% of the NTSC color standard, a high quantum yield (QY), and radiative lifetimes in the range of 1–29 ns [7].

Halide Pe-NCs are low-cost and high-quality materials which can be synthesized by facile methods, such as hot injection [7] and ligand-assisted reprecipitation (LARP) [9,10]. Due to the polar bonds in halide perovskites, the Fröhlich coupling of charge carriers to longitudinal optical (LO) phonons determines the strong electron–phonon interaction [12] which promotes the up-conversion radiative process at the expense of phonon energy and, thus, provides a phenomenon of the anti-Stokes photoluminescence (ASPL) [13]. The latter is a phonon-mediated up-conversion radiative recombination where thermal energy of the emitting material combines with an absorbed low energy photon to produce higher photon energy [14,15].

The present review describes the latest results concerning ASPL and laser cooling in Pe-NCs. First, we discuss the fundamental concept of laser cooling, then we review the available information about up-conversion mechanisms and the role of electron–phonon interactions. After that we show practical approaches for PL efficiency maximizing and laser cooling implementations.

## 2. The Concept of Optical Refrigeration

The concept of optical refrigeration (laser cooling) by one-photon up-conversion luminescence dates back to 1929 when Pringsheim proposed removing thermal vibrational energy via optical emission [14]. In a solid-state system, optical cooling is assumed to be realized under excitation with a photon together with phonon absorption, and it can result in the removal of thermal energy ΔE (see Figure 2a). The cooling rate can be controlled using an optical pump, e.g., laser irradiation [15]. The main energy flows during optical cooling are schematically shown in Figure 2b. The optical pump results in a transition from the ground electron state “0” in a solid sample to the excited ones “1” and “2”, while the coolant and cold reservoir represent the optically excited sample. The excited energy levels are coupled through phonon-assisted transitions (see Figure 2b). 

For the simplified case of a three-level system (Figure 2a), the cooling quantum efficiency, which is defined as the ratio of the cooling power to the absorbed power, can be described by the following expression [15]:(1)$$\eta =\eta_{ASPL}E_{ASPL}/E_{abs}-1$$ where $\eta_{ASPL}$ is the quantum yield of ASPL, and $E_{ASPL}$ and $E_{abs}$ are the photon energies of ASPL and the absorbed photon, respectively. The latter energies can be also expressed through the corresponding ones in the discussed three-level system (see Figure 2a). When the optical cooling in a real solid-state system with a quasi-continual spectrum of the electron energy states is considered, one should use the spectrally integrated value of $E_{ASPL}$ in Equation (1). 

In a sample with $\eta_{ASPL}\approx 1$, the cooling quantum efficiency is determined by the following energy ratio:(2)$$\eta \approx (E_{ASPL}-E_{abs})/E_{abs}=\Delta E/E_{abs}$$
where ΔE is related to the absorbed phonon energy. 

One can use Equation (1) to estimate the cooling efficiency for any given material using just the experimental spectroscopic parameters. In practice, the main condition for positive cooling efficiency corresponds to the following requirement:(3)$$\eta_{ASPL}>E_{abs}/E_{ASPL}=1\;-\;\Delta E/E_{ASPL}$$

On the one hand, Equation (3) shows that the requirement for ASPL efficiency is easily achieved when the absorbed phonon energy per one ASPL photon is larger. On the other hand, the $E_{abs}$ value should be sufficiently high to minimize the multi-phonon relaxation [15]. In the case of $\eta_{ASPL}\approx 1$, the energy removed via the ASPL process exceeds the energy added by thermalization due to nonradiative recombination, and the material finally demonstrates a net decrease in temperature.

Technological problems with material purity prevented any observations of the laser cooling effect in solids for a long time, until 1995 when the effect was observed in ytterbium-doped glass [16]. Later, rare earth metal-doped glasses were widely used for studying and demonstrating optical cooling, recently reaching temperatures as low as 91 K [17], i.e., close to the thermodynamic limit of that material system [18]. This is due to the suitable energy spacing and high fluorescence external quantum efficiency [19,20]. 

As for the efficiency of all solid-state optical cooling, direct band gap semiconductors have several advantages over materials with rare earth [16,17] and fluorescent dyes [21,22], such as high absorption and lower achievable temperature due to the quantum statistics of charge carriers [23]. This is due to the difference in the ground state populations in rare earth-doped systems and in semiconductors. For the low temperatures below 100 K in the rare earth-doped systems, the population at the top of the ground state strongly decreases due to the Boltzmann statistics of carriers, making the cooling process inefficient. To the contrary, in semiconductors, charge carriers obey the Fermi–Dirac statistics, which does not lead to the population decrease in the lower energy valence band even at temperatures close to the absolute zero. Moreover, semiconductor optical coolers could be integrated into optoelectronic devices more easily. The main problem that hindered experimental observation of laser cooling in semiconductors was low luminescence extraction efficiency [24,25]. A theory of semiconductor cooling was first proposed in Refs. [26,27], but it did not consider such important effects as luminescence trapping and red-shifting due to its reabsorption. The problem was solved in Ref. [28] where the authors described laser cooling in semiconductor structures for arbitrary external efficiency and considered for the first time the luminescence red-shift due to the reabsorption. The proposed theory was combined with the established plasma theory of semiconductor optical absorption including many-body Coulomb effects and band filling. Experimental conditions needed to attain net cooling in GaAs were derived. Finally, it was theoretically predicted to be able to reach temperatures below 10 K via the ASPL-assisted optical cooling [28,29]. Unfortunately, the optical cooling of bulk semiconductors for such low temperatures has not been demonstrated yet. For example, the bulk crystalline GaAs requires a minimum extraction efficiency of about 25% at the optimal carrier density that can be hardly achieved due to the large refractive index in GaAs [29]. 

The successful laser cooling of a semiconductor was experimentally demonstrated by Zhang and co-workers in Ref. [30] where chemical vapor deposition was used to fabricate CdS semiconductor structures, which were subwavelength in size in at least one dimension (100 nm nanobelts), to reach maximal optical extraction efficiency of the ASPL emission and then net cooling by 40 K. 

Besides III-V and II-IV compound semiconductors, the ASPL process via an efficient one-phonon-assisted up-conversion transition was also revealed in carbon nanotubes [31]. The photon up-conversion does not originate from common coherent two-photon absorption or anti-Stokes Raman processes. The observed phenomenon was attributed to an efficient excitonic up-conversion processes. The highly efficient phonon-assisted exciton up-conversion showing an energy gain much exceeding the thermal energy appeared anomalous but could be understood as a consequence of rapid exciton migration along the nanotube axis. The efficient exciton scattering by acoustic phonons or random extrinsic surface potentials spatially separated the up-converted excitons from their original localized state immediately after up-conversion and strongly reduced their probability of returning to this original state. Such unique exciton dynamics emerging in a carbon nanotube with embedded 0D-like localized states enabled efficient phonon-assisted exciton up-conversion with an energy gain exceeding the thermal energy by one order of magnitude at room temperature. While the ASPL efficiency in carbon nanotubes was far for optical cooling realization, this effect was assumed to be useful for applications in energy harvesting, optoelectronics, and bioimaging in the near-infrared (NIR) optical range [31].

To achieve efficient optical cooling in semiconductors, one should consider materials which combine strong electron–phonon coupling, high PL quantum yield (PLQY), and nanoscale size (in at least one dimension for good optical extraction of ASPL). High quality colloidal nanocrystals–quantum dots (QDs) can be synthesized with large PLQY and demonstrated well resolved ASPL [32,33,34]. In this context, halide perovskites are considered as promising candidates for semiconductor optical cooling [13,35,36,37,38,39,40]. Further, we will also discuss the restrictions in optical cooling of lead halide perovskites and propose possible ways to overcome the existing problems, among them the stability of Pe-NC-based structures and films and engineering of optimized structures of future optical cooling devices which possesses sufficient cooling efficiency for practical needs.

The bulk halide perovskites demonstrate excellent optical and electronic properties, such as strong optical absorption with a sharp absorption edge resulting from low defect density, high carrier mobility, and PLQY [41,42,43,44,45,46]. Due to the highly efficient radiative recombination with a small shift between the PL peak and the absorption edge, a phenomenon of repeated photon emission and reabsorption occurs (photon recycling). The photon recycling first observed in Ref. [47] allowed the assumption that Pe-semiconductors have an internal efficiency of almost 100% [48]. Moreover, the observed efficient ASPL [36,38] and strong electron–phonon interactions [12] have also been reported in lead halide perovskites, suggesting their potential for optical cooling.

## 3. Up-Conversion Mechanisms

In a solid-state system, the additional energy required for the ASPL excitation is usually provided by internal thermal energy in the form of phonon–lattice vibrations [15]. Pe-NCs are among the most promising materials for effective up-conversion due to their unique optical properties [7,8,9] and strong electron–phonon interaction [12,49,50,51]. Well-resolved up-conversion luminescence was observed in Pe-NCs [39,40,52,53], thin films, a thick single crystal of CH_3_NH_3_PbI_3_ [38], and two-dimensional perovskite (C_6_H_5_C_2_H_4_NH_3_)_2_PbCl_4_ [54]. 

Temperature dependent ASPL was observed in thin films of CH_3_NH_3_PbBr_3_ NCs with a size of about 7 nm deposited on glass substrates [39]. The ASPL efficiency, which was quantified by $\eta_{ASPL}$, increased 50-fold when the initial temperature of the sample increased from 100 K to room temperature. The thermally activated ASPL exhibited an activation energy of about 80 meV. At room temperature, the ASPL efficiency was about 60%, and it depended linearly on the excitation intensity. While the investigated Pe-NCs can be considered promising candidates for the demonstration of optical cooling in thin solid films; their optical properties require further improvement.

All-inorganic colloidal-combined halide CsPb(Cl/Br/I)_3_ NCs were shown to be very promising for laser cooling as they displayed well-resolved up-conversion emission via a single-photon excitation process [37-40, 51]. An efficient ASPL from CsPbBr_3_ nanostructures embedded in a host Cs_4_PbBr_6_ crystal was observed in Ref. [52]. The efficient ASPL with exceptionally large phonon-assisted energy gains of up to 8 k_B_T on all-inorganic colloidal Pe-NCs that goes beyond the maximum capability of only harvesting optical phonon modes was revealed [53]. Comparison of the up-conversion efficiency in thin film and thick crystal demonstrated up-conversion gains for thick crystal to be close to 70% that of the thin film [38]. An effective up-conversion PL from a visible self-trapped exciton to an ultraviolet free exciton in two-dimensional perovskite (C_6_H_5_C_2_H_4_NH_3_)_2_PbCl_4_ quantum wells excited by the nanosecond pulse laser excitation was studied in Ref. [54]. 

Let us now focus on the intrinsic mechanisms leading to the up-conversion PL and ASPL. Based on the experimentally observed nearly linear dependence of the ASPL intensity on pump power, the ASPL excitation was assumed, which was mediated by virtual levels or defect states [55]. The latter states are located within the band gap, and they are considered intermediate energy levels for the excited ASPL in CsPbBrI_2_ NCs [55] and CsPbBr_3_ quantum dots [56]. However, treating CsPbBr_3_ NCs with NH_4_SCN removed the trap states and increased the PLQY to near unity, which resulted in an increase in the ASPL quantum yield of such treated Pe-NCs [57]. 

Efficient PL and ASPL observed in colloidal CsPbBrI_2_ NCs under intense laser excitation tunable in the visible and NIR spectral regions (Figure 3a) were explained by considering linear and super-linear dependences of the luminescence on excitation intensity [55]. In general, the emission intensity, *I*, depended on the excitation power, *P*, in the following way: *I*∼$P^{n}$, where *n* is the number of required photons to populate the excited states. The value of *n* was approximately equal to unity when the wavelength of the laser photons was shorter than 680 nm (Figure 3b). It means that the emission occurs due to the single-photon process. For the laser photon wavelength longer than 700 nm, the PL emission involved a two-photon process. The single photon up-conversion photoluminescence was attributed to the radiative recombination of charge carriers from surface to band gap states and the two-photon emission to the carrier recombination from the conduction to the valence bands. A simplified energy diagram was proposed to illustrate the up-conversion mechanisms in CsPbBrI_2_ NCs (Figure 3c). When photon energy exceeds the band gap (*E*_ex_ > 1.978 eV), electrons in the valence band transfer to the conduction band, and the subsequent recombination of electrons and holes produces the normal PL emission (red down arrow in Figure 3c). When photon energy is slightly less than the band gap (1.824 eV < *E*_ex_ < 1.944 eV), electrons in deep trap states (yellow dashed lines) are excited and transferred to the surface state. However, the generated holes in the deep trap states are unstable and quickly transfer to the valence band. Subsequent recombination with surface state electrons produces the up-conversion luminescent emission, i.e., ASPL. With further photon energy reduction (1.784 eV < *E*_ex_ < 1.824 eV), only shallow trap states (green dashed lines) can be excited. Some incident photons are absorbed by these shallow trap states, producing single-photon up-conversion emission. However, shallow trap state density is too low to absorb all available photons. The remaining photons are absorbed by electrons in the valence band and transferred to the conduction band by a two-photon process (TP, blue arrow in Figure 3c).

Recently, self-trapped excitons were proposed as an intermediate energy level for single photon up-conversion [58]. In the case of above gap excitation, self-trapped excitons lead to the additional PL with a large Stokes shift in addition to the free exciton emission. Single trapped excitons as photoinduced transient defects are generated based on the lattice polarization and deformation caused by the above gap excitation of charge carriers [59]. 

Zhang and co-workers analyzed CsPbX_3_ NCs embedded in glasses and observed efficient ASPL, which depended on excitation wavelength and power and was explained by a phonon-assisted single-photon process [60]. In this study, femtosecond transient absorption spectra revealed that free excitons were formed with the assistance of phonons under single-photon excitation. A few ps free excitons thermally dissociated into free carriers resulted in the appearance of ASPL. The Urbach tail states derived from deformed [PbX_6_] octahedra on the outer region of CsPbX_3_ NCs were proposed to be intermediate states for the ASPL. The Urbach tail formation was explained by an effect of the surface tension of NCs and their interaction with the local environment. Excitons were supposed to be generated upon sub-gap excitation followed by localization on the Urbach states. Through electron–phonon coupling, these localized excitons changed into free excitons to generate exciton bleaching, and then thermally dissociated into free carriers accompanied by band edge bleaching, leading to an efficient ASPL.

Near-infrared ASPL was also observed in Yb/Nd co-doped CaTiO_3_ microcrystals, and it was explained by considering an up-conversion excitation via the phonon-assisted energy transfer between Yb^3+^ and Nd^3+^ ions [61]. The former ions are excited via the ^2^F_7/2_ - ^2^F_5/2_ transition by absorbing photons emitted by a 980 nm laser, and the excitation energy is subsequently transferred to Nd^3+^ ions by phonon assistance (Figure 4). The phonon energy of CaTiO_3_ is about 470 cm^−1^ [62], which can provide several phonons for energy transfer. As the temperature increases, the CaTiO_3_ lattice vibration increases and the number of phonons grows, so the efficiency of phonon-assisted energy transfer increases and ASPL intensity increases. Because the APL intensity exhibited an enhancement with temperature growth, it was proposed for optical temperature sensor applications. The proposed ASPL thermometer needed only 16 mW/mm^2^ power density, and its sensitivity was 1.52% K^−1^ [61]. 

## 4. The Role of Electron–Phonon Interaction

The strong electron–phonon interaction is usually considered an origin of the unique electrical and optical properties of both the bulk perovskites [12,50] and Pe-NCs [51,53]. It is assumed to lead the following optical phenomena: (i) enhanced ASPL [38,39,40,63] and (ii) slow hot-carrier cooling [64,65]. One of the outstanding characteristics of halide perovskites is their low optical phonon frequencies, attributed to the heavy lead ions. The linewidth of the phonon absorption is very broad at room temperatures. It means that optical phonons are overdamped due to the anharmonicity of the large phonons caused by the soft lattice nature of halide perovskites [65]. The anharmonicity influences the electron–phonon interaction characteristics [51]. The unique properties of halide perovskites are considered to be due to the large electron–phonon coupling constant, which can promote the formation of polarons [65,66]. 

Fröhlich proposed a model to describe the long-range interaction between an electron and polar optical phonons in ionic crystals or polar semiconductors [67,68]. According to this model, the strength of the electron–phonon interaction is quantified using the Fröhlich coupling constant, which is defined by the following expression [51]:(4)$$\alpha =\left(\varepsilon_{\infty }^{-1}-\varepsilon_{0}^{-1}\right)\sqrt{m_{e}^{}e^{4}/2\hbar^{3}\omega_{LO}},$$ where $\varepsilon_{\infty }^{}$ and $\varepsilon_{0}^{}$ are the high-frequency and low-frequency limits of the dielectric function, respectively; $m_{e}^{}$ is the effective mass of the electron; and $\omega_{LO}$ is the LO–phonon frequency. 

To analyze the electron–phonon interaction in halide perovskites, one can consider a steepness parameter of the temperature dependence of Urbach energy, which is expressed as follows [51]:(5)$$\sigma_{0}=s\cdot B/E_{R},$$
where *s* is a dimensionless parameter dependent on the geometrical structure of the lattice, *B* is the band width, and $E_{R}$ is the lattice relaxation energy. 

Figure 5 plots the reciprocal value of the steepness parameter as a function of the Frohlich coupling constant for different materials [51]; III–V and II–VI semiconductors with weak electron–phonon coupling are situated on the lower left side of the scheme. On the upper right side of this scheme, one can find insulators with strong electron–phonon coupling. Halide perovskites do not belong to any of these groups, and they form a separate group, which reveals the strong electron–phonon interaction and low probability of the exciton self-trapping.

The electron–phonon interaction in CH_3_NH_3_PbX_3_ lead halide perovskites was studied by observing the Landau levels and high-order excitons at weak magnetic fields when the cyclotron energy is much smaller than the LO–phonon energy [51]. It was revealed that by changing the halogen from I to Br to Cl, the electron–phonon coupling becomes stronger, which results in the larger effective mass enhancement caused by polaron formation. The presence of phonon screening of the electron-hole Coulomb interaction was revealed. It was proposed that the difference in the LO phonon energy can be due to the halogen dependence of the exciton binding energy.

Temperature dependent ASPL was observed in thin films of CH_3_NH_3_PbBr_3_ NCs with a size of about 7 nm deposited on glass substrates [39]. The ASPL efficiency, which was quantified by $\eta_{ASPL}$, was found to increase 50-fold when the initial temperature of the sample increased from 100 K to room temperature. The thermally activated ASPL exhibited an activation energy of about 80 meV. At room temperature, the ASPL efficiency was about 60% and it depended linearly on the excitation intensity. While the investigated Pe-NCs are considered promising candidates for the demonstration of optical cooling in thin solid films, their optical properties are required for further optimization.

Size-dependent room temperature ASPL in CH_3_NH_3_PbX_3_ Pe-NCs with mean sizes varying from about 6 nm to 120 nm was investigated, and the observed enhancement of ASPL intensity with the reduction in the mean NC size was explained by the strong electron–phonon interaction and size-dependent energy bandgap [40,63]. It was assumed that excitons in an ensemble of larger nanocrystals (smaller bandgap) could be formed either without phonon participation or through one-photon absorption accompanied by simultaneous participation of several phonons to provide the momentum conservation in the direct gap semiconductor (Figure 6a). On the one hand, the radiative decays of such excitons provide either normal Stokes PL or the ASPL emission with relatively low efficiency. On the other hand, the ASPL emission in small Pe-NCs can be excited via simultaneous one-photon and one-phonon absorption (Figure 6b) The latter occurs in the regime of the strong quantum confinement in Pe-NCs that results in the breaking of the quasi-momentum selection rule [63] because it is favorable in small semiconductor NCs [69].

## 5. PL Efficiency Enhancement

Because high efficiency of the ASPL process is required for optical cooling applications (see Equation (2)), let us discuss the possible approaches for its enhancement in Pe-NCs and the main practically achieved results on this issue. While one ASPL quantum can take away energy ΔE, which corresponds to the energy difference between incident and emitted photons and accounts typically for several tens of meV, one act of the nonradiative photon absorptions gives an energy in the order of 1 eV [15]. Thus, to remove the energy in total, it is necessary to have a high QY, i.e., around 95% and higher. Ha and co-authors [36] calculated the theoretical threshold of QY for optical cooling in the case of CH_3_NH_3_PbI_3_ NCs (Figure 7).

Recent studies in the field of Pe-NCs have shown that their optical properties are mostly determined by the quality of synthesized NCs. Structural defects acting as trap states are the key factor that limit not just QYs but also the stability of those Pe-NCs [70,71]. Removing surface trap states can enhance radiative recombination pathways and homogenize the exciton concentration, suppressing nonradiative recombination [71,72]. In addition to defect trap states and nonradiative recombination processes, the following are the known limiting factors of QY: weak exciton binding, slow cooling of long-lived hot carriers, deep-level defects, and ion migration [70,71].

Various approaches to overcome the abovementioned limitations have been reported [71,72,73,74,75]. They can be divided into three main categories: (i) nanoscaling; (ii) synthesis methods and perovskite compound modifications; (iii) surface engineering via surface ligand development.

Decreasing perovskite particle sizes down to the nanoscale (or decreasing perovskite dimensionality to 2D, 0D) leads to the exciton’s confinement. The more an exciton is confined to the higher binding energy, the shorter its diffusion length. Increasing the exciton binding energy and decreasing the exciton diffusion length improve the PL efficiency of Pe-NCs [74]. According to Kim and co-workers [75], Pe-NCs of 11–27 nm sizes demonstrate the highest QY, while further size deduction leads to a decrease in the QY values. The decrease can be attributed to the increase in trap-assisted recombination of excitons at surface traps due to the increased surface-to-volume ratio because the trap-assisted recombination is mainly related to the nonradiative recombination [75].

Synthesis method modifications and post-synthesis doping of Pe-NCs are mainly aimed to reduce the density of halide vacancies, which are the well-recognized defects represented by the emergence of metallic lead in convenient Pe-semiconductors [76]. These metallic sites act as non-radiative recombination centers for electron trapping facilitating nonradiative mechanisms [77]. Cho and co-workers [78] prevented the formation of metallic lead (Pb) atoms through a small increase in methylammonium bromide’s (MABr) molar proportion during MAPbBr_3_ PNCs synthesis. Thiocyanate salts, such as NH_4_SCN or NaSCN, were used by Koscher and co-workers [79] to remove metallic Pb and to enhance QY of Pe-NCs from 92% to 99% for fresh samples and from 63% to 100% for aged ones (stored for a few months).

Liu and co-workers [80] showed that a high room temperature PL QY of up to 100% can be obtained in CsPbI_3_ perovskite quantum dots (QDs), signifying the achievement of almost complete elimination of the trapping defects. Their improved synthetic protocol that involves introducing the organo–lead compound trioctylphosphine–PbI_2_ (TOP–PbI_2_) as the reactive precursor also led to a significantly improved stability for the resulting CsPbI_3_ QD solutions.

Partial replacement of Pb atoms in the perovskite structure was proposed to improve and modify PL properties of Pe-NCs. Moreover, this provides better commercialization possibilities of perovskites due to a decrease in Pb contamination. The substitution of Pb^2+^ by Mn^2+^ causes orange light emission and improves the radiative carrier recombination dynamics. A near-unity PLQY was achieved for a Mn:Pb molar ratio of 3:1 in combination with CdCl_2_ post-synthesis treatment [81]. Partial substitution of Pb by Sr in iodine red-emitting Pe-NCs significantly increases the stability of their PL properties and also increases QY for a long period of time [82] (Figure 8).

Yong and co-workers [83] reported a general strategy for the synthesis of all-inorganic violet-emitting Pe-NCs with near-unity QY through engineering of the local order of the lattice using nickel ion doping.

Yb^3+^ doping of CsPbCl_3_ Pe-NCs allowed an additional mechanism to enhance PL efficiency due quantum cutting, i.e., when one high-energy photon is absorbed and then two low-energy photons are generated. Milstein and co-authors [84] demonstrated that excitonic perovskite PL is almost suppressed in this case, and PL from Yb^3+^ states with PLQY significantly above 100% can be achieved (up to 170% from the theoretical limit of 200%).

Ligand engineering approaches for the PL QY enhancement of Pe-NC aim to improve surface passivation for better stability [76,77]; prevent agglomeration, which leads to loss of nanoscale advantages [70,71,72,73,74]; improve homogeneity [63,76]; and reduce the isolating effect of ligands for LED applications [85,86]. For example, Di Stasio and co-workers [85] proposed a facile route to enhance the photophysical properties of FAPbBr_3_ NCs via synthesis with short-chain capping ligands such as octylamine and octanoic acid, i.e., the simultaneous additions of oleic acid and PbBr_2_ to the redispersed material in toluene, prior to purification with methyl acetate. The oleic acid addition removes Pe-NC aggregates, while PbBr_2_ provides an excess of Br^−^ anions to passivate the halide defects and strengthen the ligand binding on the NCs’ surfaces. The resulting QY > 95% of the exciton PL was achieved for a thin film of CsPbBr_3_ NCs [86].

The surface coverage of Pe-NCs can be improved by mixing those NCs with polymers, such as polymethylmethacrylate (PMMA) [87], polyvinylpyrrolidone [88], and polystyrene [89]. The PMMA matrix, having a low dielectric permittivity, increases the exciton binding energy in Pe-NCs, and the photoluminescence QY of the Pe-NC film increases up to 92%, which is explained by an effect of the dielectric confinement for excitons in Pe-NCs [87]. Furthermore, the polymer coatings improve long-term storage of Pe-NCs without significant losses in their QY [87,88,89].

To improve ligand binding and to minimize degradation of the PL QY during long-term storage, a post-synthetic treatment with didodecyldimethylammonium bromide (DDAB) has been proposed [90,91,92]. Due to the strong binding with negatively charged sites in Pe-NCs, DDAB forms a hydrophobic monolayer, increasing the long-term stability of the structural and luminescent properties of the ligand-passivated Pe-NCs. For example, Bodnarchuk and co-authors [90] used DDAB and DDAB+PbBr_2_ post-synthetic treatment of CsPbBr_3_ colloidal solutions, which had a low initial PL QY below 70%. The post-synthetic treatment of Pe-NCs resulted in the PL QY increasing up to 100% (Figure 9). Additionally, it was shown that the surface modification by DDAB does not affect PL peak position and does not cause its broadening [90,91].

## 6. Optical Cooling Implementation

The recent results that have been discussed demonstrate that if crystalline perovskites and colloidal Pe-NCs can be prepared with low trap densities, then high external PL quantum efficiency is achieved. Thus, the necessary condition for optical cooling realization [15] is practically achievable (see Equation (2)); its implementation is currently an urgent task. Indeed, the high PL efficiencies of bulk and low-dimensional perovskites and Pe-NCs under and above band gap excitations are also advantageous for efficient ASPL excitation in these materials under phonon-assisted one-photon absorption processes [13,36,37,38,39,40,60,63].

A new strategic approach for engineering the optical properties of perovskite nanoparticles to achieve optical cooling was proposed in Ref. [93]. The authors theoretically demonstrated the optimization of ASPL efficiency in MAPbI_3_ nanoparticles via excitation of Mie resonances both at emission and absorption wavelengths. The optimized theoretical photo-induced temperature decrease was achieved for a hybrid halide perovskite 530 nm nanoparticle on a glass substrate by more than 100 K under cw illumination at 980 nm and intensities being about 7 × 10^6^ W/cm^2^. The numerical and analytical modeling revealed that the highest cooling efficiencies for halide perovskite nanoparticles correspond to the excitation of magnetic type Mie modes (magnetic octupole at the emission and magnetic quadrupole at absorption).

Ref. [36] presented bight examples of the realization of net cooling by 23.0 K in micrometer thick 3D CH_3_NH_3_PbI_3_ (MAPBI_3_) and by 58.7 K in exfoliated 2D (C_6_H_5_C_2_H_4_NH_3_)_2_PbI_4_ (PhEPbI_4_) perovskite crystals directly from room temperature. Three-dimensional platelets were grown using vapor phase synthesis, and 2D samples were exfoliated from a bulk crystal obtained via solution synthesis. Figure 10a,c demonstrate the Stokes PL at 296 K and the corresponding absorption spectra for the 3D and 2D abovementioned perovskites, respectively. The PL peaks are at 770 nm and 529 nm for the 3D and 2D samples, respectively. Towards the band tail, the absorption decreases and reaches zero, demonstrating that there should not be any substantial phonon-assisted ASPL beyond this wavelength. The possible cooling regime corresponds to the situation, when the excitation photon energy is less than mean PL emission (the green area in Figure 10a,c). The intensity on ASPL versus pump power is shown in Figure 10b,d. The inset in Figure 10b shows that for the 3D case, below a certain excitation power, the ASPL intensity linearly depends on laser power. Such behavior corresponds to the domination of the phonon-assisted up-conversion. The deviation from the linear dependence above this power is possibly due to the sample degradation at high laser powers. For 2D perovskite, the linear dependence of ASPL intensity on pump power was revealed in the entire range of pump intensities, revealing that up-conversion is a phonon-assisted process (see Figure 10d).

To measure the cooling effect of the crystals, the pump–probe luminescence thermometry technique, which was previously used in rare earth-doped materials and semiconductors, was adopted [94]. For the 3D perovskite platelet, the PL redshift was revealed during pumping at 785 nm, indicating a cooling process. In contrast, 760 nm pumping leads to a blue-shifted band edge, thus indicating a heating process. After switching off the pump lasers, the photoluminescence spectra returned to their original position indicating that the cooling/warming cycle is reversible. It was demonstrated that the 3D perovskite crystal could be cooled by a maximum of 23 K from room temperature if pumped by 785 nm with a power of 0.66 mW. The normalized cooling power density is shown in Figure 11a.

A maximum cooling effect of 35 K/mW around 780–785 nm was revealed [36], which strongly exceeds the results obtained previously for CdS nanobelts [30]. Theoretical calculations were performed using the theory discussed in Refs. [25,26]. The same results were obtained for 2D perovskites. The maximum cooling by 58.7 K directly from room temperature was achieved with 565 nm laser excitation. The authors of Ref. [32] also demonstrated an actual perovskite optical cooler in which CdSe nanobelts coupled to a 2D perovskite were cooled by 28 K from room temperature.

The first demonstration of optical cooling driven by colloidal semiconductor nanocrystal up-conversion was performed in Ref. [37]. In this study, CsPbBr_3_ NCs were shown to cool down during the up-conversion of 532 nm continuous-wave (cw) laser excitation; the resultant ASPL spectrum was collected continuously or at regular intervals during the course of the experiment. The Raman thermometric analysis of the substrate on which the nanocrystals were deposited further verified a decrease in the local temperature by at least 25 K during optical pumping. The Arrhenius behavior of the ASPL yield was used to estimate the change in temperature of Pe-NCs during below-gap excitation.

According to the results of Ref. [37], the optical cooling rate and achievable temperature were dependent on the excitation laser fluence. The temperature change of the environment around Pe-NCs was analyzed by monitoring the anti-Stokes to Stokes Raman scattering ratio of a silicon substrate on which Pe-NCs were deposited. This analytical thermometry technique measured the temperature-dependent phonon mode population of silicon to obtain the local temperature of the substrate. During below-gap excitation, the ASPL spectra changed over time, decreasing in intensity as a function of excitation fluence and demonstrating a red-shift of the spectral position of the PL peak. During ASPL, each up-conversion event can be considered a cooling cycle that removes thermal energy by decreasing the population of the phonon modes of the Pe-NCs. The decrease in the phonon mode population reduces ASPL yield and removes an amount of thermal energy. This photoinduced thermal deactivation is manifested in a fluence dependence of the ASPL’s rate of signal decay observed by the authors experimentally. Higher laser fluence removed a larger amount of thermal energy causing a faster decay of the ASPL intensity (see Figure 12).

Figure 13 shows temperature transients during the optical cooling experiment performed in Ref. [37]. The sample was pumped below gap with a fluence of 300 W/cm^2^ such that thermal energy was removed faster than it was replaced by the surrounding environment of Pe-NCs in the optical spot. The temperature dropped exponentially and reached approximately 10°C in 2 min. The excitation fluence was then decreased to 30 W/cm^2^, below the fluence threshold necessary to overcome the heat flux from the environment into the nanocrystals. When the fluence was decreased below the cooling threshold, the nanocrystal temperature started to increase. The laser was blocked for 5 min to demonstrate the continued reversal of the decay in ASPL intensity. After approximately 20 min, nanoparticles demonstrated room temperature (1400 s). Additionally, the authors studied the cooling of a silicon substrate. Both the temperature dependent yield of ASPL and the anti-Stokes and Stokes Raman scattering of CsPbBr_3_ NCs demonstrated the net decrease in temperature. The final temperature of the silicon substrate was −1.7°C, and the final temperature of Pe-NCs was estimated to be −5°C. The observed cooling is comparable in magnitude with the results obtained for 2D perovskites [36].

## 7. Perspectives and Conclusions

The investigations of various inorganic and metalorganic perovskites as well low-dimensional Pe-structures and Pe-NCs discussed previously demonstrate efficient phonon-assisted anti-Stokes photoluminescence under relatively low-intensive optical excitation, which is commonly realized by cw laser irradiation in the visible and NIR spectral ranges. The high ASPL quantum yield, which should be close to unity, and the strong coupling of the exciting photons and phonons, are necessary conditions to achieve net optical cooling. While the fundamental issues of both the luminescence mechanism and electron–phonon interaction in perovskites are nearly understood, an enhancement of the electron–phonon coupling and ASPL efficiency in Pe-NCs are still actual tasks for optical cooling applications. 

Recently, several approaches and strategies for the improvement of ASPL efficiency and stability of Pe-NC-based structures and films are proposed and realized. Preparation and employment of surface-passivated Pe-NCs with narrow size distribution seem to be a promising strategy for the practical realization of all-solid optical cooling. Another actual task consists of engineering an optimized structure for a future optical cooling device (all-solid optical refrigerator) which possesses sufficient cooling efficiency for practical needs. However, considering the impressive results of recent years in improving the parameters of semiconductor perovskites for solar cells and light-emitting devices, it can be expected that low-dimensional Pe-semiconductors and Pe-NCs can also be further optimized for applications in optical cooling technologies. Among the possible applications of all-solid-state laser cooling are infrared cameras, ultra-stable laser resonators, spectrometers and optical detectors, low-noise optical amplifiers which require the absence of vibrations, and other existing and future photonic devices; their performance of is hard to achieve using traditional cooling technologies.

## Figures and Tables

**Figure 1 nanomaterials-13-01833-f001:**
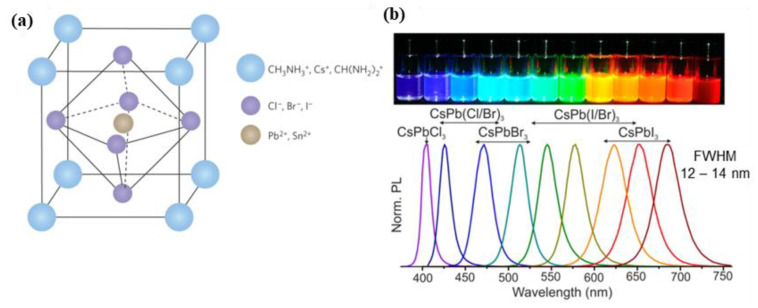
(**a**) Halide perovskite crystal ABX_3_ structure where large blue circles correspond to the first cations as CH_3_NH_3_^+^, Cs^+^, and CH(NH_2_)_2_^+^; violet circles depict the anions as Cl^−^, Br^−^, and I^−^; and brown circles depict the second cation as Pb^2+^ and Sn^2+^. Reprinted from Ref. [8]; (**b**) representative images of the luminescent colloidal solutions of CsPbX_3_ NCs and their PL spectra. Reprinted from Ref. [7].

**Figure 2 nanomaterials-13-01833-f002:**
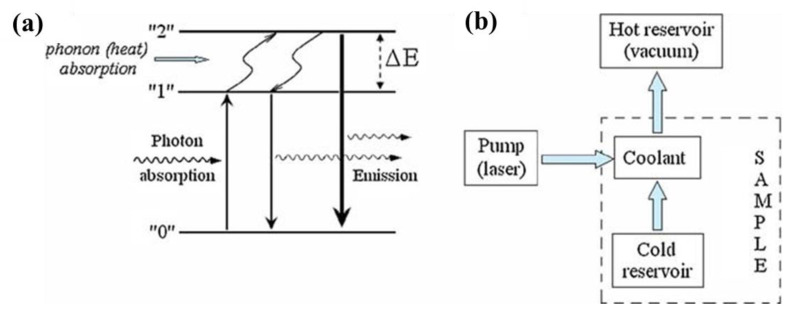
(**a**) Schematic illustration of the PL excitation and emission (thin vertical arrows) and ASPL (thick vertical arrow); (**b**) schematic illustration of the optical refrigeration. Adopted from Ref. [15].

**Figure 3 nanomaterials-13-01833-f003:**
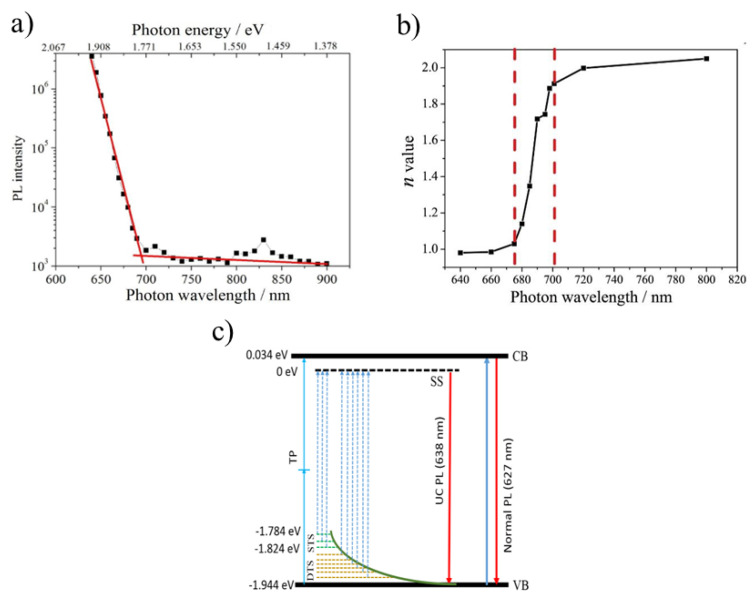
(**a**) PL intensity of CsPbBrI_2_ NCs excited by 650–900 nm wavelength incident photons; (**b**) *n* values of the *I~P^n^* dependence for CsPbBrI_2_ NCs excited by different wavelength laser; (**c**) Simplified energy diagram and up-conversion PL. CB, conduction band; VB, valence band; SS, surface state; STS, shallow trap state; DTS, deep trap state; TP, two-photon process. Reprinted from Ref. [55].

**Figure 4 nanomaterials-13-01833-f004:**
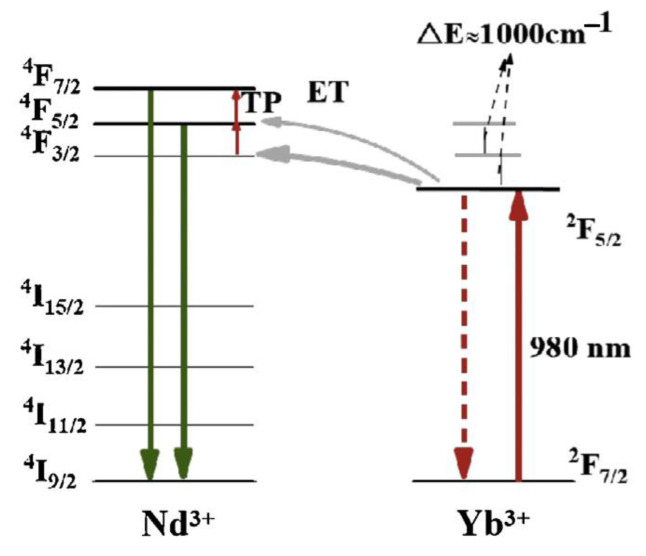
The energy level diagram of Nd^3+^ and Yb^3+^ ions in CaTiO_3_:Yb/Nd and the corresponding anti-Stokes process under 980 nm excitation. TP and ET denote processes of the thermal population and energy transfer, respectively. Reprinted from Ref. [61].

**Figure 5 nanomaterials-13-01833-f005:**
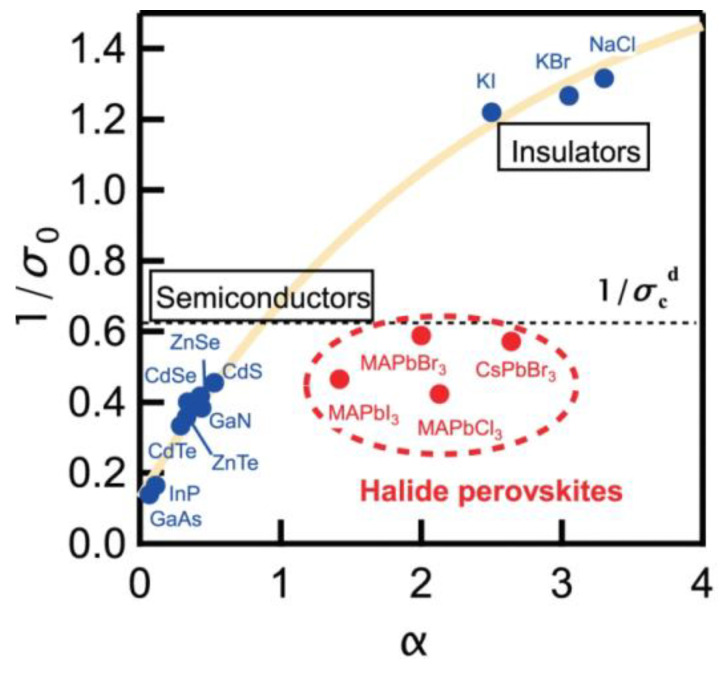
The relation between the Fröhlich coupling constant *α* and the inverse steepness parameter for various direct-gap materials. The dotted line is a theoretically predicted threshold value of the inverse steepness for the formation of self-trapped excitons. Reprinted from Ref. [51].

**Figure 6 nanomaterials-13-01833-f006:**
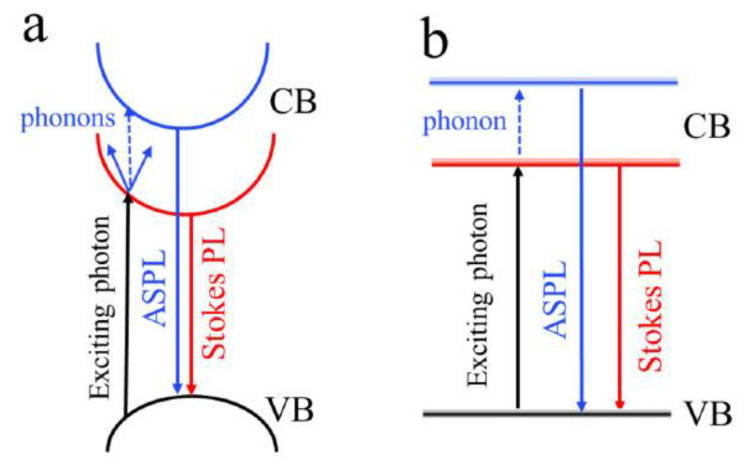
Energy diagrams of Stokes PL and ASPL generation in (**a**) large size CH_3_NH_3_PbBr_3_ PNCs and (**b**) small size CH_3_NH_3_PbBr_3_ PNCs. Reprinted from Ref. [63].

**Figure 7 nanomaterials-13-01833-f007:**
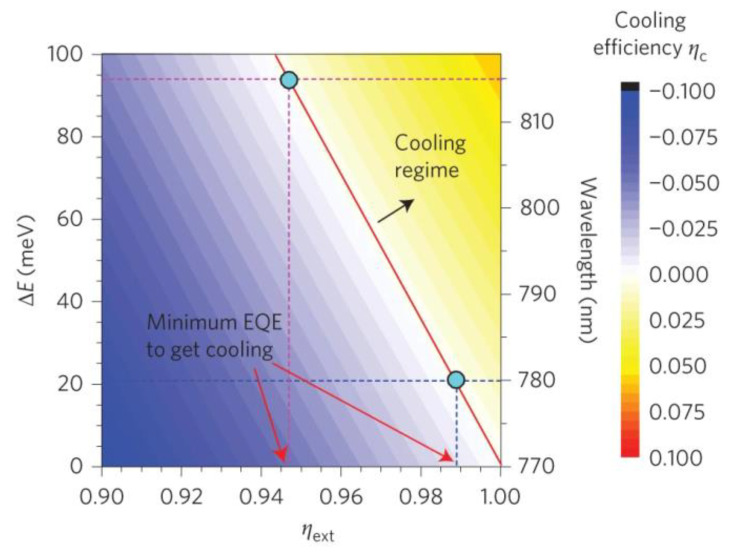
Calculated cooling efficiency as a function of external quantum efficiency and energy difference between the excitation photon and emission photon energies for CH_3_NH_3_PbI_3_ NCs. Reprinted from Ref. [36].

**Figure 8 nanomaterials-13-01833-f008:**
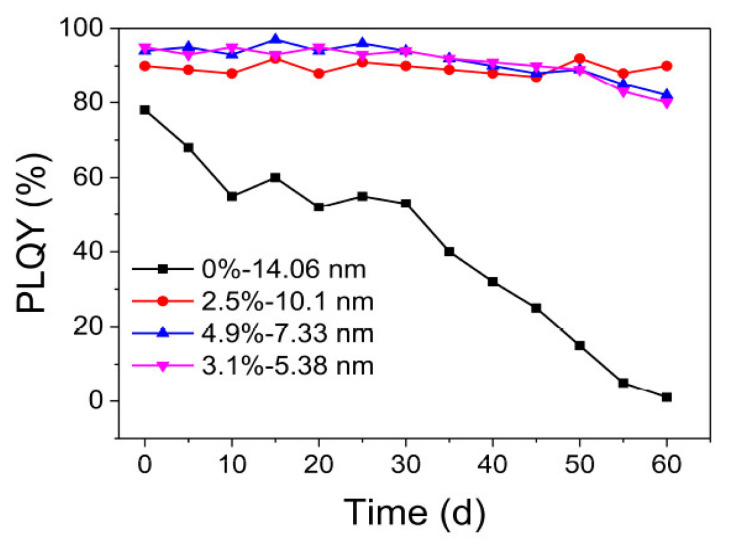
PLQY values as a function of aged days for unsubstituted and Sr^2+^-substituted CsPbI_3_ quantum dot solutions. Reprinted from Ref. [82].

**Figure 9 nanomaterials-13-01833-f009:**
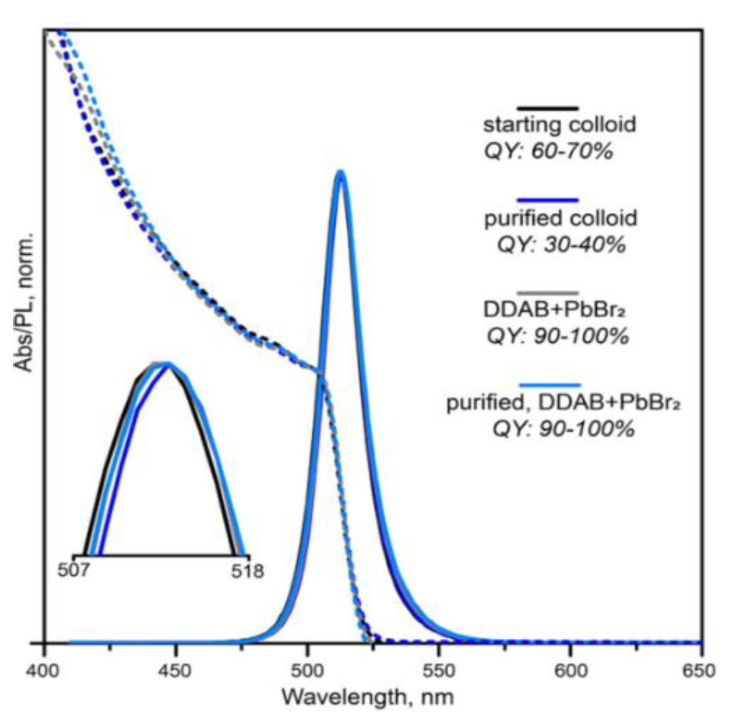
Comparison of the steady-state absorption and PL spectra for the starting colloid of CsPbBr_3_ NCs and the same colloid subjected to several treatments. Reprinted from Ref. [90].

**Figure 10 nanomaterials-13-01833-f010:**
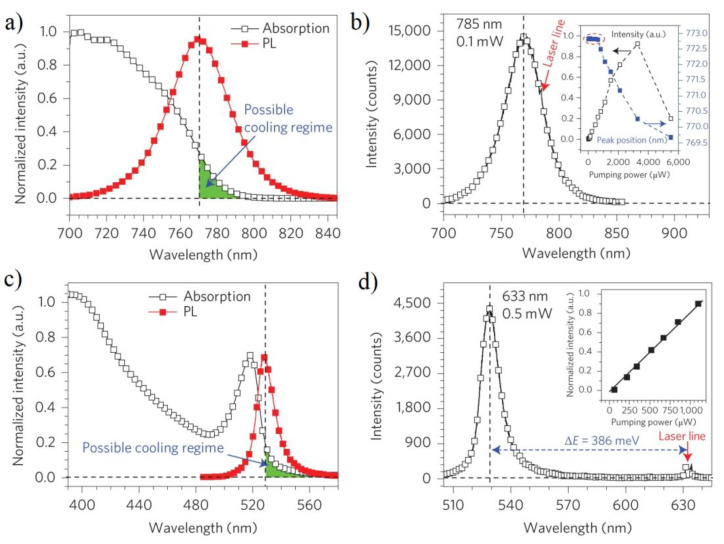
PL/ASPL and optical absorption spectra of 3D (2D) perovskites are shown in panels (**a**–**d**), respectively. (**a**,**c**) PL and the absorption spectra at 296 K. The green area indicates possible cooling regime. (**b**,**d**) ASPL spectra at 296 K. The insets to (**b**,**d**) show the PL intensity dependences on excitation power, the inset to (**b**) also shows the peak position as a function of pump power. Reprinted from Ref. [36].

**Figure 11 nanomaterials-13-01833-f011:**
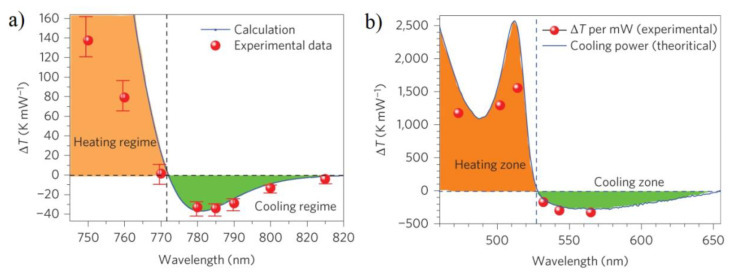
Net laser cooling for 3D (panel (**a**)) and 2D (panel (**b**)) perovskites. Panels demonstrate measured maximum normalized cooling power density (filled circles) and theoretical calculations within the theory presented in [25] (solid curve). Reprinted from Ref. [36].

**Figure 12 nanomaterials-13-01833-f012:**
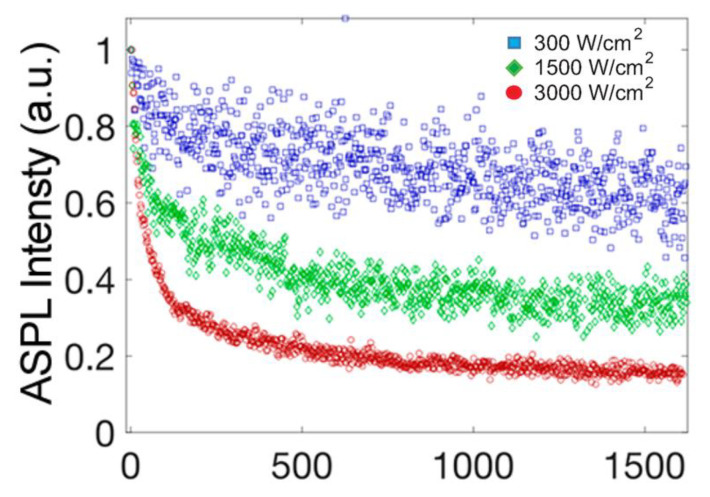
ASPL intensity of CsPbBr_3_ NPs over time during below-gap (532 nm) excitation. Blue squares, green diamonds, and red circles correspond to excitation fluences of 300, 1500, and 3000 W/cm^2^, respectively. Reprinted from Ref. [37].

**Figure 13 nanomaterials-13-01833-f013:**
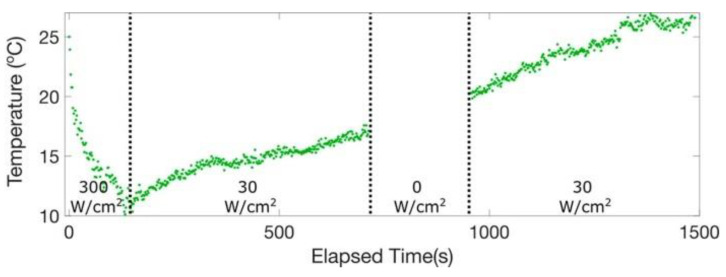
Temperature transients of CsPbBr_3_ NCs film over time during below gap excitation. The temperature of the sample was estimated using ASPL intensity. Reprinted from Ref. [37].

## Data Availability

Not applicable.

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
