# Peer review of "Anti-Stokes Photoluminescence in Halide Perovskite Nanocrystals: From Understanding the Mechanism towards Application in Fully Solid-State Optical Cooling"

_nanomaterials, 2023, doi:10.3390/nano13121833_

Round 1
Reviewer 1 Report
The manuscript combines the last published results devoted to the anti-Stokes photoluminescence in perovskite nanocrystals. I consider this work as a important for the community, hence I recommend its publication in Nanomaterials in the current form.
Author Response
We thank the Reviewer for the high evaluation of our paper.
Reviewer 2 Report
Halide perovskite is considered as a promising candidate for semiconductor optical cooling. In this review, the basic principle of ASPL is derived and analyzed in detail. The high ASPL and the strong coupling of the exciting photons and phonons are necessary conditions to achieve net optical cooling. At the same time different types of literatures are listed to discuss the influence of the size, distribution and surface passivation of PE-NCs on the efficiency of ASPL. At the same time, the relationship between the light excitation energy and temperature is also discussed.
This work is written well, and the study is comprehensive. I can recommend its publication after the minor revisions.
Some points need to be addressed:
1. Line60, there is a syntax error, Missing subject “then we review”
2. Fig.1a, the names of each ion can be placed below the image
3. Line 82, there is a format error. “ASPL-induced”
4. Line 138-142, why carbon nanotubes cannot be well applied and optically cooled, and what are the reasons that affect the ASPL efficiency of carbon nanotubes?
5. Line 149-157, halide perovskite is considered to be a promising candidate for semiconductor optical cooling. But is there any factors restrict the development of optical cooling of lead halide perovskite? What should be the development direction of optical cooling of lead halide perovskite in the future.
6. Fig3, the coordinate axis text and picture size in Figure 3 should be consistent,
7. Line236, there is a problem with the format of ion labels
8. Line 279-280 there is a format error :‘the steepness, ?0’s
9. Fig12, there is no legend for different color points in the figure. The author should mark different colors in the figure corresponding to different excitation fluxes
10. Line 541-542 there is a format in the first sentence.
11. Line542-561, optical cooling technology and all-solid optical cooling are considered a promising candidate for semiconductor optical cooling. Can the author list several commercial practical application cases, or what problems need to be solved in the industrialization of halide perovskite semiconductor cooler in the future.
Minor editing of English language required
Author Response
We thank the Reviewer for the evaluation of our manuscript and for the helpful comments and suggestions.
Below we give the list changes and our answers (A) to the Reviewers questions (Q).
Q1. Line 60, there is a syntax error, Missing subject “then we review”.
A1. We made all the necessary changes in the text of the manuscript.
Q2. Fig.1a, the names of each ion can be placed below the image.
A2. We modified the caption in Fig.1 and added names of each ion depicted in panel a.
Q3. Line 82, there is a format error. “ASPL-induced”.
A3. We made the corresponding corrections in the text of the manuscript.
Q4. Line 138-142, why carbon nanotubes cannot be well applied and optically cooled, and what are the reasons that affect the ASPL efficiency of carbon nanotubes?
A4. We added a short discussion to the text of the manuscript to clarify the reasons that affect ASPL efficiency of carbon nanotubes (lines 143-154).
Q5. Line 149-157, halide perovskite is considered to be a promising candidate for semiconductor optical cooling. But is there any factors restrict the development of optical cooling of lead halide perovskite? What should be the development direction of optical cooling of lead halide perovskite in the future.
A5. We added discussion to the text of the revised manuscript (lines 162-166). We also would like to mention that in the Section Conclusion and Perspectives more detailed discussion has been also added.
Q6. Fig3, the coordinate axis text and picture size in Figure 3 should be consistent.
A6. We tried to do our best to make the text and the picture sizes consistent.
Q7. Line236, there is a problem with the format of ion labels.
A7. We eliminated the problem from the text (line 254 in the revised manuscript).
Q8. Line 279-280 there is a format error :‘the steepness, ?0’s.
A8. We eliminated the problem from the text (lines 297-298 in the revised manuscript).
Q9. Fig12, there is no legend for different color points in the figure. The author should mark different colors in the figure corresponding to different excitation fluxes.
A9. We made changes in Fig.12 and added the legend.
Q10. Line 541-542 there is a format in the first sentence.
A10. We formatted the text in a correct way.
Q11. Line542-561, optical cooling technology and all-solid optical cooling are considered a promising candidate for semiconductor optical cooling. Can the author list several commercial practical application cases, or what problems need to be solved in the industrialization of halide perovskite semiconductor cooler in the future.
A11. We added a discussion concerning the possible applications of laser cooling (lines 578-582 in the revised manuscript).
Reviewer 3 Report
Anti-Stokes photoluminescence (ASPL) is very significant in the study of perovskite (Pe) nanocrystals, which makes it potentially valuable in the field of optical all-solid cooling or optical refrigeration. This review comprehensively expounds the mechanism and the influencing factors of ASPL, which is of certain reading value. However, before the review can be accepted, the authors need to complete the following minor revisions.
1. The author involved 11 keywords in the Abstract, but many of them are not mentioned in the corresponding exposition content. It is hoped that the author could simplify the keywords based on the summary content.
2. The title of the paper shows the research on perovskite nanocrystals, but from the overall description of the review, the author almost involves the research content of halide perovskite nanocrystals. Would it be more appropriate for the author to change perovskite nanocrystals to halide perovskite nanocrystals in the title?
Minor modifications of English language required.
Author Response
1. We have made changes to the list of keywords to make them shorter and relevant.
2. We agree with the Reviewer. We added “halide” to the title of the manuscript.